# How are age-related differences in sleep quality associated with health outcomes? An epidemiological investigation in a UK cohort of 2406 adults

Andrew Gadie,[1] Meredith Shafto,[2] Yue Leng,[3] Rogier A Kievit,[1] Cam-CAN[4]

[1]MRC Cognition and Brain Sciences Unit, Cambridge, UK
[2]Department of Psychology, University of Cambridge, Cambridge, UK
[3]Department of Psychiatry, University of California, San Francisco, California, USA
[4]Cambridge Centre for Ageing and Neuroscience (Cam-CAN), University of Cambridge and MRC Cognition and Brain Sciences Unit, Cambridge, UK

**Correspondence to**
Dr Rogier A Kievit;
rogier.kievit@mrc-cbu.cam.ac.uk

## ABSTRACT

**Objectives** To examine age-related differences in self-reported sleep quality and their associations with health outcomes across four domains: physical health, cognitive health, mental health and neural health.

**Setting** Cambridge Centre for Ageing and Neuroscience (Cam-CAN) is a cohort study in East Anglia/England, which collected self-reported health and lifestyle questions as well as a range of objective measures from healthy adults.

**Participants** 2406 healthy adults (age 18–98) answered questions about their sleep quality (Pittsburgh Sleep Quality Index (PSQI)) and measures of physical, cognitive, mental and neural health. A subset of 641 individuals provided measures of brain structure.

**Main outcome measures** PSQI scores of sleep and scores across tests within the four domains of health. Latent class analysis (LCA) is used to identify sleep types across the lifespan. Bayesian regressions quantify the presence, and absence, of relationships between sleep quality and health measures.

**Results** Better self-reported sleep is generally associated with better health outcomes, strongly so for mental health, moderately for cognitive and physical health, but not for sleep quality and neural health. LCA identified four sleep types: 'good sleepers' (68.1%, most frequent in middle age), 'inefficient sleepers' (14.01%, most frequent in old age), 'delayed sleepers' (9.28%, most frequent in young adults) and 'poor sleepers' (8.5%, most frequent in old age). There is little evidence for interactions between sleep quality and age on health outcomes. Finally, we observe U-shaped associations between sleep duration and mental health (depression and anxiety) as well as self-reported general health, such that both short and long sleep were associated with poorer outcomes.

**Conclusions** Lifespan changes in sleep quality are multifaceted and not captured well by summary measures, but instead should be viewed as as partially independent symptoms that vary in prevalence across the lifespan. Better self-reported sleep is associated with better health outcomes, and the strength of these associations differs across health domains. Notably, we do not observe associations between self-reported sleep quality and white matter.

### Strengths and limitations of this study

► Broad phenotypic assessment of healthy ageing across multiple health domains.
► Advanced analytic techniques (ie, latent class analysis regression) allows new insights.
► A uniquely large neuroimaging sample combined with Bayesian inference allows for quantification of evidence for the null hypothesis.
► Subjective sleep measures may have drawbacks in older samples.
► Cross-sectional data precludes modelling of within subject changes.

## BACKGROUND

Sleep is a fundamental human behaviour, with humans spending almost a third of their lives asleep. Regular and sufficient sleep has been shown to benefit human physiology through a number of different routes, ranging from consolidation of memories[1] to removal of free radicals[2] and neurotoxic waste.[3] Sleep patterns are known to change across the lifespan in various ways, including decreases in quantity and quality of sleep,[4] with up to 50% of older adults report difficulties initiating and/or maintaining sleep.[5] A meta-analysis of over 65 studies reflecting 3577 subjects across the lifespan reported a complex pattern of changes, including an increase of stage 1 but a decrease of stage 2 sleep in old age, as well as a decrease in rapid eye movement (REM) sleep.[6] An epidemiological investigation of self-reported sleep in older adults observed marked sex differences in age-related sleep changes, with females more likely to report disturbed sleep onset but men reporting night-time awakenings.[7] Other findings are age-related physiological changes in the alignment of homeostatic and circadian rhythms,[8] decreases in sleep

efficiency,[9] the amount of slow-wave sleep and an increase in daytime napping.[10] Importantly, interruption and loss of sleep have been shown to have wide-ranging adverse effects on health,[11] leaving open the possibility that age-related changes in sleep patterns and quality may contribute to well-documented age-related declines in various health domains.

In the current study, we examine self-reported sleep habits in a large, population-based cohort Cambridge Centre for Ageing and Neuroscience (Cam-CAN[12]). We relate sleep measures to measures of health across four health domains: cognitive, brain health, physical and mental health. Our goal is to quantify and compare the associations between typical age-related changes in sleep quality and a range of measures of health measures that commonly decline in later life. We assess sleep using a self-reported measure of sleep quality, the Pittsburgh Sleep Quality Index (PSQI).[13] The PSQI has good psychometric properties[14] and has been shown to correlate reliably with diseases of ageing and mortality.[15–17] Although polysomnography[18] is commonly considered the gold standard of sleep quality measurement, it is often prohibitively challenging to employ in large samples. A recent direct comparison of sleep measures[19] suggests that although subjective sleep measures (such as PSQI) may have certain drawbacks in older samples, they also capture complementary aspects of sleep quality not fully captured by polysomnography. Moreover, collecting self-report sleep quality data in a large, deeply phenotyped cohort offers several additional benefits.

By using a population cohort of healthy adults and studying a range of health outcomes in the same population, we can circumvent challenges associated with studying clinical populations and provide new insights. First and foremost, by investigating associations between sleep and outcomes across multiple health domains in the same sample, we can make direct comparisons of the relative magnitude of these effects. Second, larger samples allow us to generate precise effect size estimates, as well as adduce in favour of the null hypothesis. Third, we investigate the associations between sleep quality and neural health in a uniquely large healthy population. Previous investigations of the consequences of poor sleep, especially on neural health, generally focused on clinical populations such as those suffering from insomnia.[20 21] Although such studies are crucial for understanding pathology, the demographic idiosyncrasies and often modest sample sizes of these approaches make it hard to generalise to healthy, community-dwelling lifespan populations. Moreover, most studies that study age-related changes or differences focus on (very) old age, while far less is known about young and middle-aged adults.[6] For these reasons, our focus on a healthy, multimodal lifespan cohort is likely to yield novel insights into the subtle changes in sleep quality across the lifespan.

We will focus on three questions within each health domain. First, is there a relationship between sleep quality and health? Second, does the strength and nature of this relationship change when age is included as a covariate? Third, does the strength and nature of the relationship change across the lifespan? We will examine these questions across each of the four health domains.

## METHODS
### Sample
A cohort of 2544[12] was recruited as part of the population-based Cam-CAN cohort (www.cam-can.com), drawn from the general population via Primary Care Trust's (PCT) lists within the Cambridge City (UK) area; 10 520 invitation letters were sent between 2010 and 2012, and willing participants were invited to have an interview conducted in their home, with questions on health, lifestyle demographics and core cognitive assessments. Sample size was chosen to allow for 100 participants per decile in further acquisition stages, giving sufficient power to separate age-related change from other sources of individual variation. For additional details of the project protocol, see Shafto et al[12] and Taylor et al[22] and for further details of the Cam-CAN dataset visit http://www.mrc-cbu.cam.ac.uk/datasets/camcan/. A further subset of participants who were MRI compatible with no serious cognitive impairment participated in a neuroimaging session[22] between 2011 and 2013. Participants included were native English speakers, had normal or corrected to normal vision and hearing, and scored 25 or higher on the Mini-Mental State Examination (MMSE).[23] Note that other, more stringent cut-offs are sometimes employed to screen for premorbid dementia, such as a score of 88 or higher in the Addenbrooke's Cognitive Examination Revised (ACE-R).[24] For the sake of comprehensiveness, we repeated our analyses using this more stringent cut-off (ACE-R >88), but observed no noteworthy differences in our findings, so we only report the findings based on the MMSE exclusion criteria . Ethical approval for the study was obtained from the Cambridgeshire 2 (now East of England-Cambridge Central) Research Ethics Committee (reference: 10/H0308/50). Participants gave written informed consent. The raw data and analysis code are available on signing a data sharing request form (see http://www.mrc-cbu.cam.ac.uk/datasets/camcan/ for more detail).

### Variables
#### Sleep measures
Sleep quality was assessed using the PSQI, a well-validated self-report questionnaire[13 19] designed to assist in the diagnosis of sleep disorders. The questions concern sleep patterns, habits and lifestyle questions, grouped into seven components, each yielding a score ranging from 0 (good sleep/no problems) to 3 (poor sleep/severe problems), that are commonly summed to a PSQI total score ranging between 0 and 21, with higher scores reflecting poorer sleep quality.

### Health measures
#### Cognitive health

A number of studies have found associations between poor sleep and cognitive decline, including in elderly populations. Poor sleep affects cognitive abilities such as executive functions[25] and learning and memory processes,[26] whereas short-term pharmaceutical interventions such as administration of melatonin improve both sleep quality and cognitive performance.[27 28] Recent work[29] concluded that 'maintaining good sleep quality, at least in young, adulthood and middle age, promotes better cognitive functioning and serves to protect against age-related cognitive declines'. As sleep may affect various aspects of cognition differently,[30] we include measures that cover a range of cognitive domains including memory, reasoning, response speed and verbal fluency, as well a measure of general cognition (see table 1 and Shafto et al[12] for more details).

#### Neural health

Previous research suggests that individuals with a severe disruption of sleep are significantly more likely to exhibit signs of poor neural health.[20 31] Specifically, previous studies have observed decreased white matter health in clinical populations suffering from conditions such as chronic insomnia,[21] obstructive sleep apnoea,[32 33] excessively long sleep in patients with diabetes[34] and REM sleep behaviour disorder.[35] Many of these studies focus on white matter hyperintensities (WMHs), a measure of the total volume or number of regions showing low-level neural pathology (although some study grey matter, eg, Macey et al[36] and Sexton et al[37]). WMHs are often used as a clinical marker, as longitudinal increases in WMHs are associated with increased risk of stroke, dementia and death[38] and are more prevalent in patients with pathological sleep problems.[33 34] However, use of this metric in clinical cohorts largely leaves open the question of the impact of sleep quality on neural (white matter) health in non-clinical, healthy populations. To address this question, we use a more general indicator of white matter neural health; fractional anisotropy (FA). FA is associated with white matter integrity and myelination.[39 40] We use FA as recent evidence suggests that WMHs represent the extremes (foci) of white matter damage and that FA is able to capture the full continuum of white matter integrity.[41] For more information regarding the precise white matter pipeline, see Shafto et al[12], Taylor et al[22] and Kievit et al.[42]

#### Physical health

Sleep quality is also an important marker for physical health, with poorer sleep being associated with conditions such as obesity, diabetes mellitus,[43] overall health[11 44] and increased all-cause mortality.[45 46] We focus on a set of variables that capture three types of health domains commonly associated with poor sleep: cardiovascular health measured by pulse, systolic and diastolic blood pressure,[47] self-reported health, both in general and for the past 12 months[48] and body mass index.[49]

#### Mental health

Previous work has found that disruptions of sleep quality are a central symptom of forms of psychopathology such as major depressive disorder, including both hypersomnia and insomnia,[44 50] and earlier episodes of insomnia greatly increased the risk of later episodes of major depression.[51] Kaneita et al[52] found a U-shaped association between sleep and depression, such that individuals regularly sleeping less than 6 hours, or more than 8 hours, were more likely to be depressed. Both depression[53] and anxiety[54 55] are commonly associated with sleep problems. To capture these dimensions, we used both scales of the Hospital Anxiety and Depression Scale (HADS),[56] a widely used and standardised questionnaire that captures self-reported frequency and intensity of anxiety and depression symptoms.

## STATISTICAL ANALYSES

We examined whether self-reported sleep patterns change across the lifespan, both for the PSQI sum score and for each of the seven PSQI components. We then examined the relationships between the sleep quality and the four health domains in three ways. First, simple regression of the health outcome on sleep variables to determine evidence for association between poor sleep quality and poor health outcomes. Second, we included age as a covariate. Finally, we included a (standard normal rescaled) continuous interaction term to examine whether there is evidence for a changing relationship between sleep and outcomes across the lifespan.

For all regressions, we used a default Bayesian approach advocated by Liang et al[57], Rouder and Morey,[58] Wagenmakers[59] and Wetzels et al[60] which avoids several well-documented issues with p-values,[59] allows for quantification of null effects and decreases the risk of multiple comparison problems.[61] Bayesian regressions allows us to symmetrically quantify evidence in favour of, or against, some substantive model as compared with a baseline (eg, null) model. This evidential strength is expressed as a Bayes factor,[62] which can be interpreted as the relative likelihood of one model versus another given the data and a certain prior expectation. A Bayes factor of, for example, 7 in favour of a simple linear regression model suggests that the data are seven times *more likely* under that model than an intercept only model for a given prior (for an empirical comparison of p-values and Bayes factors, see Wetzels et al[60]). A heuristic summary of evidential interpretation can be seen in figure 1.

We report log Bayes factors for (very) large effects and regular Bayes factors for smaller effects. To compute Bayes factors, we used default Bayes factor approach for model selection[57 58] in the package BayesFactor[63] using the open source software package R.[64] As previous papers report associations between sleep and outcomes ranging from absent to considerable in size, we used the default, symmetric Cauchy prior with width $\frac{\sqrt{2}}{2}$ which translates to a 50% confidence that the true effect will lie between

**Table 1** Description of health variables across each of four domains (cognitive, neural, physical, mental)

| Health domain | Task and description | Variable | Descriptives | References |
|---|---|---|---|---|
| Cognitive | Story Recall Immediate: participants hear a short story and are asked to recall as accurately as possible. | Recall manually scored for similarity and precision (min=0, max=24) | n=2379, M=13.14, SD=4.66, range=0–24 | 92 |
| Cognitive | Story Recall Delayed: same as above but recall after 30 min delay | Recall manually scored for similarity and precision (min=0, max=24) | n=2366, M=11.47, SD=4.92, range=0–24 | 92 |
| Cognitive | Letter Fluency (phonemic fluency): participants have 1 min to generate as many words as possible beginning with the letter 'p'. | Total words generated (min=0,max=30) | n=2360, M=25.38, SD=3.96, range=0–30 | 92 |
| Cognitive | Animal Fluency (semantic fluency): participants have 1 min to generate as many words as possible in the category 'animals'. | Total words generated (min=0,max=30) | n=2346, M=25.85, SD=4.47, range=0–30 | 92 |
| Cognitive | Cattell Culture Fair: test of fluid reasoning using four subtests (series completions, odd one out, matrices and topology) | Total correct summed across four subtests (min=0, max=46) | n=658, M=31.8, SD=6.79, range=11–44 | 93 |
| Cognitive | Simple reaction time: speed in a simple reaction time task | 1/response time in seconds | n=657, M=0.37, SD=0.08, range=0.24–0.93 | 12 |
| Cognitive | Addenbrooke's Cognitive Examination, Revised: screening test for dementia using seven subtests (orientation, attention and concentration, memory, fluency, language, visuospatial abilities, perceptual abilities) | Performance on multiple tests converted to min=0, max=100 range | n=2406, M=89.25, SD=13.4, range=0–100 | 24 |
| Neural | White matter health: measure of tract integrity using fractional anisotropy | Fractional anisotropy (min=0, max=1, averaged across 10 tracts) | n=641, M=0.5, SD=0.03, range=0.3–0.56 | 73 |
| Physical | Self-reported health, in general: participants use a four-point scale to respond to the prompt "Would you say for someone of your age, your own health in general is…" | Score from 1=excellent to 4=poor | n=2404, M=2.02, SD=0.79, range=1–3[1–3] | 93 |
| Physical | Self-reported health, last 12 months: participants use a three-point scale to respond to the prompt "Over the last twelve months would you say your health has on the whole been…" | Score from 1=good to 3=poor | n=2398, M=1.46, SD=0.71, range=1–3[1–3] | 93 |
| Physical | Systolic blood pressure | Mean systolic blood pressure in mm Hg, averaged across three consecutive measurements | n=577, M=120.11, SD=17, range=78.5–186 | |
| Physical | Diastolic blood pressure | Mean diastolic blood pressure in mm Hg, averaged across three consecutive measurements | n=577, M=73.14, SD=10.48, range=49–115.5 | |
| Physical | Resting pulse | Mean pulse in beats per minute, averaged across three consecutive measurements | n=578, M=65.69, SD=10.5, range=40–110.5 | |

**Table 1** Continued

| Health domain | Task and description | Variable | Descriptives | References |
|---|---|---|---|---|
| Physical | Body Mass Index (BMI) | (weight in kg)/(height in m$^2$) | n=584, M=25.77, SD=4.59, range=16.75–48.32 | 94 |
| Mental health | Anxiety Subscale (Hospital Anxiety and Depression Scale (HADS)): participant's response to seven questions about anxiety-related behaviours | Seven questions rated on 0–3 scale ('often' to 'very seldom') (min=0, max=21) | n=2391, M=5.13, SD=3.31, range=0–17 | 56 |
| Mental health | Depression Subscale (HADS): participant's response to seven questions about depression-related behaviours | Seven questions rated on 0–3 scale ('often' to 'very seldom') (min=0, max=21) | n=2373, M=3.32, SD=2.91, range=0–14 | |

For each variable, details are given including a description of the task it is derived from, relevant citations, a brief definition and descriptive statistics.

| Bayes Factor BF10 | Log BF10 | Tileplot colour | Description (Jeffreys, 1961) |
|---|---|---|---|
| >100 | >4.6 | | Extreme evidence for H1 |
| 30 – 100 | 3.4 – 4-6 | | Very strong evidence for H1 |
| 10 – 30 | 2.3 – 3.4 | | Strong evidence for H1 |
| 3 – 10 | 1.098 – 2.3 | | Substantial evidence for H1 |
| 1 – 3 | 1 – 1.098 | | Anecdotal evidence for H1 |
| 1 | 0 | | No evidence either way |
| 1/3 – 1 | -1.098 – -1 | | Anecdotal evidence for H0 |
| 1/3 - 1/10 | -2.3 – -1.098 | | Substantial evidence for H0 |
| 1/10 - 1/30 | -3.4 – -2.3 | | Strong evidence for H0 |
| 1/30 - 1/100 | -4.6 – - 3.4 | | Very strong evidence for H0 |
| <1/100 | < -4.6 | | Extreme evidence for H0 |

**Figure 1** Descriptive interpretation of Bayes factors.

−0.707 and 0.707. Prior to further analysis, scores on all outcomes were transformed to a standard normal distribution, and any scores exceeding a z-score of 4 or −4 were recoded as missing (aggregate percentage outliers across the four health domains: cognitive, 0.41%; mental, 0.16%; neural, 0.37%; physical, 0.031%).

## RESULTS

### Age-related differences in sleep quality

First, we examined sleep changes across the lifespan by examining age-related differences in the PSQI sum score (n=2178, M=5.16, SD=3.35, range=0–19). Regressing the PSQI global score on age (see online supplementary figure 1) showed evidence for a positive relationship across the lifespan ($logBF_{10}$=10.45). This suggests that on the whole, sleep quality decreases across the lifespan (note that higher PSQI scores correspond to worse sleep). Although we observed strong statistical evidence for an age-related difference ('extreme' according to Jeffreys[62]), age explained only 1.23% of the variance in the PSQI total score. Next, we regressed each of the seven components on age in the same manner. In online supplementary figure 2, we see that age has varying and specific effects on different aspects of sleep quality and did not worsen uniformly across the lifespan. For example, we observed moderate evidence that sleep latency did not change across the lifespan (sleep latency, $BF_{01}$=9.25, in favour of the null), sleep quality showed no evidence for either change or stasis ($BF_{10}$=1.63) and one sleep component, daytime dysfunction, improved slightly across the lifespan ($BF_{10}$=7.03). The strongest age-related decline is that of efficiency, showing an r-squared of 6.6%.

Finally, we entered all seven components into a Bayesian multiple regression simultaneously to examine to what extent they could, together, predict age. The best model included every component except sleep latency ($logBF_{10}$=142.71). Interestingly, this model explained 13.66% of the variance in age, compared with 1.23% for the PSQI total score and 6.6% for the strongest single component (efficiency). This shows that lifespan changes in self-reported sleep are heterogeneous and partially independent and that specific patterns and components need to be taken into account simultaneously to fully understand age-related differences in sleep quality. These

finding shows that neither the PSQI sum score nor the sleep components in isolation fully capture differences in sleep quality across the lifespan.

The analysis above suggests that conceptualising 'poor sleep' as a single dimension does not reflect the subtleties in lifespan changes—an often computed sumscore changes little across the lifespan, whereas the totality of sleep symptoms shows far stronger, and more subtle, patterns. To better elucidate individual differences in sleep quality, we next use latent class analysis (LCA).[65] This technique will allow us examine individual differences in sleep quality across the lifespan in more detail than afforded by simple linear regressions: rather than examining continuous variation in sleep components, LCA classifies individuals into different *sleep types*, each associated with a distinct profile of 'sleep symptoms'. If there are specific constellations of sleep problems across individuals, we can quantify and visualise such sleep types.

To analyse the data in this manner, we binarised the responses on each component into 'good' (0 or 1) or 'poor' (2 or 3). Our measures of PSQI symptoms straddle the border between continuous and categorical— although some are fully continuous (eg, sleep latency) others are less so. For instance, although scored on a range of four several of the scales (such as Subjective Sleep quality) have implicitly binary response options of 'very good' and 'fairly good' on the one hand and 'fairly bad' and 'very bad' on the other. As analytical work in psychometrics[66] suggests that Likert-like graded scales can be treated as continuous only from five ordinal categories upwards, by fitting an LCA we are erring on the side of caution (although a latent profile analysis would likely give similar results). Note that although our analysis divides individuals into discrete classes with specific profiles, it is still possible to examine the conditional response likelihood of responding 'yes' to each symptom as a continuous metric (between 0 and 1) that reflects the nature of the association between the class and the outcome. By modelling sleep 'types', we hope to illustrate the complex patterns in a more intelligible manner— notably, doing so allows us to examine whether the likelihood of belonging to any sleep 'type' changes as a function of age.

Next, we examined evidence for distinct sleep types using a set of possible models (varying from 2 to 6 sleep types). We found that the four-class solution gives the best solution, according to the Bayesian Information Criterion[67] (BIC for four classes=11 825.65, lowest BIC for other solutions=11 884.92 (five classes) (with 50 repetitions per class, at 5000 maximum iterations)). Next, we inspected the nature of the sleep types, the prevalence of each 'sleep type' in the population and whether the likelihood of belonging to a certain sleep type changes across the lifespan. See figure 2 for the component profiles of the four sleep types identified.

Class 1, 'good sleepers', makes up 68.1% of participants. Their sleep profile is shown in figure 2A, top left, and is characterised by a low probability of responding

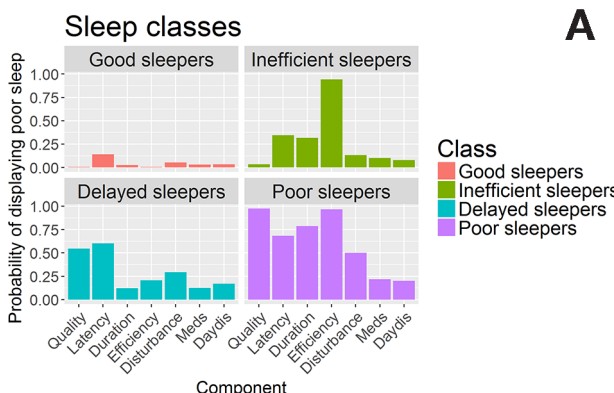

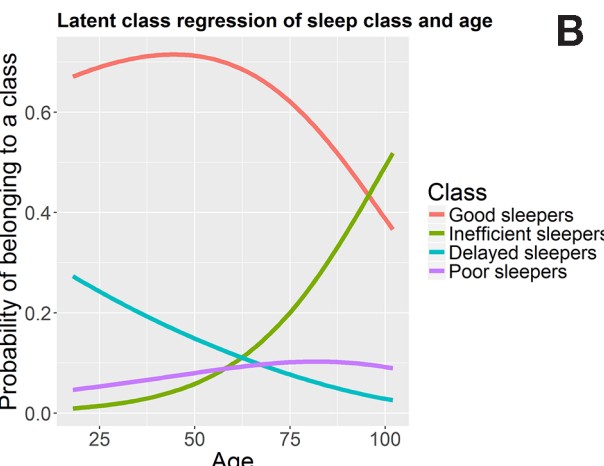

**Figure 2** Latent class analysis. Panel (A) shows the sleep quality profiles for each of the four classes. Panel (B) shows the conditional probability of belonging to each class across the lifespan.

'poor' to any of the sleep components. Class 2, 'inefficient sleepers', makes up 14.01% of the participants and is characterised by poor sleep efficiency: members of this group uniformly (100%) report poor sleep efficiency, despite relatively low prevalence of other sleep problems, as seen in figure 2A, top right. Class 3, 'delayed sleepers' seen in the bottom left of figure 2A, makes up 9.28% of the participants: characterised by modestly poor sleep across the board, but a relatively high probability of poor scores on sleep latency (59%), sleep quality (51%) and sleep disturbance (31%). Finally, class 4, 'poor sleepers', makes up 8.5% of the participants, shown bottom right in figure 2A. Their responses to any of the seven sleep components are likely to be 'poor' or 'very poor', almost universally so for 'sleep quality' (94%) and 'sleep efficiency' (97.7%).

Next, we included age as a covariate (simultaneously including a covariate is known as *latent class regression* or concomitant-variable latent class models).[68] This analysis, visualised in figure 2B, shows that the probability of membership of each classes compared with the reference class (good sleepers) changes significantly across

the lifespan for each of the classes (class 2 vs class 1: beta/SE=0.05/0.00681, t=7.611; class 3 vs class 1: beta/SE=−0.01948/0.0055, t=−3.54; class 4 vs class 1: beta/SE 0.01269/0.00478, t=2.655), for more details on generalised logit coefficients, see Linzer and Lewis.[65] The frequency of class 1 (good sleepers) peaks in middle to late adulthood, dropping increasingly quickly after age 50. Class 2 (inefficient sleepers) are relatively rare in younger individuals, but the prevalence increases rapidly in individuals over age 50. On the other hand, class 3 (delayed sleepers) shows a steady decrease in the probability of an individual showing this profile across the lifespan, suggesting that this specific pattern of poor sleep is more commonly associated with younger adults. Finally, the proportion of class 4 (poor sleepers) members increases only slightly across the lifespan. Together, the LCA provides additional evidence that the PSQI sum score as an indicator of sleep quality does not fully capture the subtleties of age-related differences. Age-related changes in sleep patterns are characterised by specific, clustered patterns of sleep problems that cannot be adequately characterised by summation of the component scores. The above analyses show how both a summary measure and individual measures of sleep quality change across the lifespan. Next, we examined the relationships between sleep quality measures (seven components and the global PSQI score) and health variables (specific variables across four domains, as shown in table 1).

## Sleep, health domains and age

### Cognitive health

First, we examined the relationships between sleep quality and seven measures of cognitive health (see table 1 for details). We visualise our findings using tileplots.[69] Each cell shows the numeric effect size (r-squared, 0–100) of the bivariate association between a sleep component and a health outcome, colour coded by the statistical evidence for a relationship using the Bayes factor. If the parameter estimate is positive, the r-squared value has the symbol '+' added (note the interpretation depends on the nature of the variable, cf. table 1).

As can be seen in online supplementary figure 3, several relationships exist between measures of cognitive health and measures of sleep quality. However, these results attenuate in a multiple regression model including age as shown in figure 3.

The cognitive abilities most strongly associated with poor sleep are a measure of general cognitive health, ACE-R and a test of verbal phonemic fluency. Two patterns emerged. First, the strongest predictor across the simple and multiple regressions was for the PSQI total score. Tentatively, this suggests that a cumulative index of sleep problems, rather than any specific pattern of poor sleep, is the biggest risk factor for poorer cognitive performance. Second, after controlling for age, the most strongly affected cognitive measure is phonemic fluency, the ability to generate name as many different words as possible starting with a given letter within a minute.

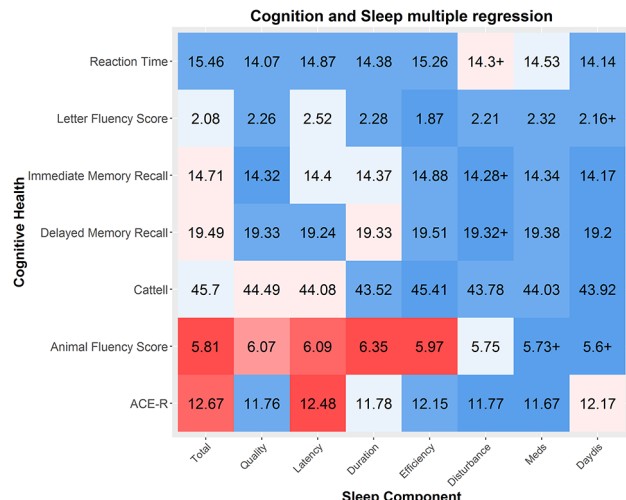

**Figure 3** Multiple regressions between sleep components and cognitive health. The strength of the effect is colour coded by Bayes factor and the effect size is shown as r-squared (as a percentage out of 100). Sample varies across components and measures due to varying missingness. Cattell and reaction time were measured only in the imaging cohort: mean N=648, N=11.11. Sample sizes for five other domains are similar (mean n=2300.25, SD=65.57). ACE-R, Addenbrooke's CognitiveExamination Revised.

Verbal fluency is commonly used as a neuropsychological test.[70] Previous work suggests that it depends on both the ability to cluster (generating words within a semantic cluster) and to switch (switching between categories) and is especially vulnerable to frontal and temporal lobe damage (with specific regions dependant on either a semantic or phonemic task[71]). Although modest in size, our findings suggests this task, dependent on multiple executive processes, is particularly affected by poor sleep quality.[72] The second strongest association was with the ACE-R, a general cognitive test battery similar in style and content to the MMSE.Lilittle evidence for interactions with age was observed (mean $logBF_{10}$=−2.08, see online supplementary figure 4), suggesting that the negative associations between sleep and cognitive performance are a constant feature across the lifespan, rather than specifically in elderly individuals. Together, this suggests that poor sleep quality is modestly but consistently associated with poorer general cognitive performance across the lifespan, most strongly with semantic fluency.

### Neural health

Using diffusion tensor imaging, we estimated a general index of white matter integrity in 10 tracts[73] (shown in online Supplementary figure 5), by taking the average FA in each white matter region of interest (ROI) (see Kievit et al[42] for more information). We used the data from a subsample of 641 individuals (age M=54.87, range 18.48–88.96) who were scanned in a 3T MRI scanner (for more details regarding the pipeline, sequence and processing steps, see Taylor et al[22] and Kievit et al[42])<-this reference is a bit of a mess - reference 74 and 42 are the same paper, this reference should only cite the watershed paper once

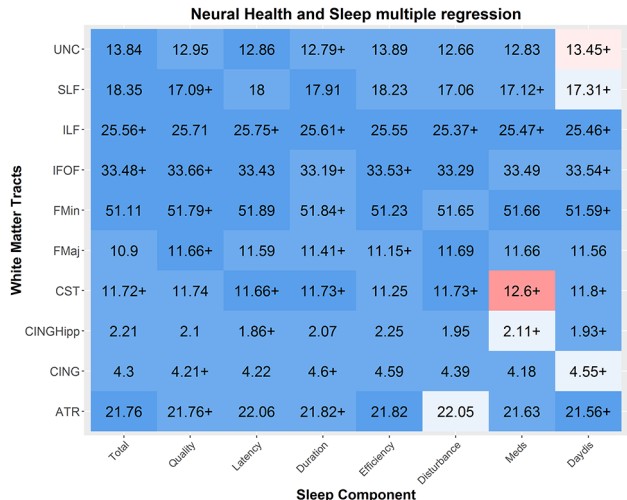

**Figure 4** Multiple regressions between sleep components and neural health. Each cell represents the relationship between a sleep component and the mean neural health in a given tract as index by fractional anisotrophy. Numbers represented in r-squared. Strong associations are observed between measures of Sleep Medication usage and multiple tracts, along with sporadic associations between other components and tracts. White matter tracts abbreviations: uncinated fasciculus (UNC), superior longitudinal fasciculus (SLF), inferior longitudinal fasciculus (ILF), inferior fronto-occipital fasciculus (IFOF), forceps minor (FMin), forceps major (FMaj), cerebrospinal tract (CST), the ventral cingulate gyrus (CINGHipp), the dorsal cingulate gyrus (CING) and the anterior thalamic radiations (ATR). N varies slightly across components due to varying missingness (N mean=631.325, SD=10.32).

(reference 42). Regressing neural white matter ROI's on sleep quality, we find several small effects, with the strongest associations between sleep efficiency and neural health (see online supplementary figure 6). All effects are such that poorer sleep is associated with poorer neural health, apart from a small effect in the opposite direction for uncinate and daytime dysfunction (BF$_{10}$=6.20). However, when age is included as a covariate, the negative associations between sleep quality and white matter health are attenuated virtually to zero (figure 4, mean/median BF$_{10}$=0.18/.10), with Bayes factors providing strong evidence for the lack of associations between sleep quality and white matter integrity. One exception was observed: the use of sleep medication is associated with *better* neural health in the corticospinal tract, a region previously found to be affected by pathological sleep problems such as sleep apnoea.[33] However, this effect is very small (BF$_{10}$=3.24) given the magnitude of the sample and the range of comparisons, so should be interpreted with caution.

Finally, we tested for any interactions by including a mean-scaled interaction term (sleep*age, see online supplementary figure 7). This analysis found evidence for a significant interaction, between the superior longitudinal fasciculus (SLF) and sleep medication (BF$_{10}$=13.77), such that better neural health in the SLF was associated

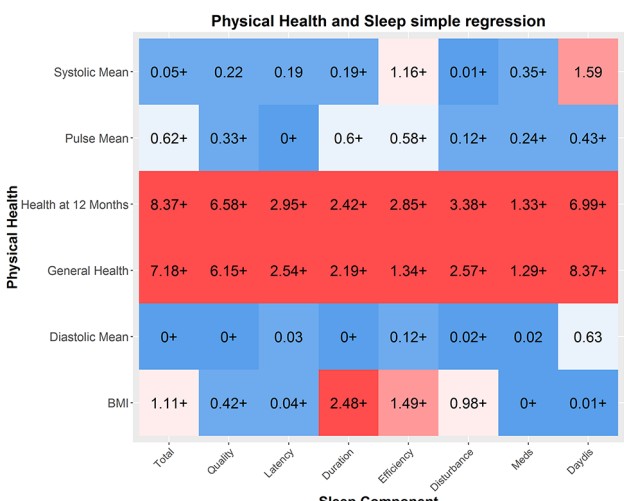

**Figure 5** Physical health and sleep quality. Numbers represent r-squared. Strong associations between general indices of health and sleep quality are found, and several modest relationships with BMI and sleep quality. Self-reported health (12 months and general) were measured in the full cohort (mean=2315.37, SD=66.29), the other indicators were measured in the imaging cohort only (mean=569.87, SD=11.16). BMI, body mass index.

with the use of sleep medication more strongly in older adults. Together, these findings suggest that in general, once age is taken into account, self-reported sleep problems in a non-clinical sample are *not* associated with poorer neural health, although there is some evidence for a modest associations between better neural health in specific tracts and the use of sleep medication in the elderly.

### Physical health

Next, we examined whether sleep quality is associated with physical health. Figure 5 shows the simple regressions between sleep quality and physical health. Strong associations were found between poor overall sleep (PSQI sum score) and poor self-reported health, both in general (logBF$_{10}$=77.51) and even more strongly for health in the past 12 months (logBF$_{10}$=91.25). This may be because poorer sleep, across all components, directly affects general physical health[43 74] or because people subjectively experience sleep quality as a fundamental part of overall general health. A second association was between BMI and poor sleep quality, most strongly poor duration (logBF$_{10}$=4.69).

This not only replicates previous findings but is in line with an increasing body of evidence that suggests that short sleep duration causes metabolic changes, which in turn increases the risk of both diabetes mellitus and obesity.[43 75 76] Next, we examined whether these effects were attenuated once age was included. We show that although the relationships are slightly weaker, the overall pattern remains (see online supplementary figure 8), suggesting that these associations are not merely co-occurrences across the lifespan. Our findings suggest self-reported sleep quality, especially sleep duration, is

related to differences in physical health outcomes in a healthy sample.

Finally, there was evidence of a single interaction with age (see online supplementary figure 9). Although poor sleep duration was associated with *higher* diastolic blood pressure in younger adults, it was associated with *lower* diastolic blood pressure in older individuals ($BF_{10}$=8.53). This may reflect the fact that diastolic blood pressure is related to cardiovascular health in a different way across the lifespan, although given the small effect size it should be interpreted with caution.

### Mental health

Finally, we examined the relationship between sleep quality and mental health, as measured by the HADS.[56] One benefit of the HADS in this context is that, unlike some other definitions (eg, the Diagnostic and Statistical Manual of Mental Disorders-V), sleep quality is not an integral (scored) symptom of these dimensions. As shown in online supplementary figure 10, there are very strong relationships between all aspects of sleep quality and measures of both anxiety and depression. The strongest predictors of depression are daytime dysfunction ($\log BF_{10}$=245.9, $r^2$=19.26%), followed by the overall sleep score ($\log BF_{10}$=170.5, $r^2$=14.92%) and sleep quality ($\log BF_{10}$=106.8, $r^2$=8.9%). The effects size for anxiety was comparable but slightly smaller in magnitude. When age is included as a covariate, the relationships remained virtually unchanged (see online supplementary figure 11), suggesting that these relationships are present throughout across the lifespan. These findings replicate and extend previous work, suggesting that sleep quality is strongly associated with both anxiety and depression across the lifespan.

Finally, we examined a model with an interaction term (see online supplementary figure 12). Most prominently, we found interactions with age in the relationship between HADS anxiety and the PSQI total, such that for the relationship between anxiety and overall sleep quality

is stronger in younger adults ($BF_{10}$=4.6, see figure 6). Together, our findings show that poor sleep quality is consistently, strongly and stably associated with poorer mental health across the adult lifespan.

### Non-linear associations between sleep and health outcomes

In the above analyses, we focused on linear associations between symptoms and health outcomes. However, for one aspect of sleep, namely, sleep duration (in hours), evidence exists that these associations are likely to be non-linear, such that both shorter and longer than average sleep are associated with poorer health outcomes (eg, Grandner and Drummond,[77] Kaneita et al[78] and Grandner et al[79]). This is echoed in clinical criteria for depression, which commonly include both hypersomnia and hyposomnia as 'sleep disruption' symptoms. In other words, both too much and too little sleep are suboptimal. To examine whether we observe evidence for non-linearities, we examined the relationship between raw scores on sleep duration (in hours, not transformed to PSQI norms) and health outcomes across the four domains. If the association between sleep and outcomes is indeed U-shaped (or inverted U, depending on the scale), then a Bayesian regression would prefer the less parsimonious model that includes the quadratic term. We observed no non-linear associations between any neural or cognitive health variables. We found strong evidence for a quadratic (subscript q) over a linear (subscript l) associations between sleep duration and HADS anxiety ($\log BF_{ql}$=19.98), even more strongly so with HADS depression ($\log BF_{ql}$=26.41, figure 7A shows the strongest curvilinear association, namely with depression). We found a similar U-shaped curve with general health ($BF_{ql}$=277.81) and self-reported health over the last 12 months ($BF_{ql}$=887.6), the former shown in figure 7B. Together, these analyses support previous conclusions that some (although not all) poorer health outcomes can be associated with both too much and too little sleep.

### DISCUSSION

In this study, we report on the associations between age-related differences in sleep quality and health outcomes in a large, age-heterogeneous sample of community-dwelling adults of the Cam-CAN cohort. We found that sleep quality generally decreases across the lifespan, most strongly for sleep efficiency. However age-related changes in sleep patterns are complex and multifaceted, so we used LCA to identify 'sleep types' associated with specific sleep quality profiles. We found that younger adults are more likely than older adults to display a pattern of sleep problems characterised by poor sleep quality and longer sleep latency, whereas older adults are more likely to display inefficient sleeping, characterised by long periods spent in bed while not asleep. Moreover, the probability of being a 'good' sleeper, unaffected by any adverse sleep symptoms, decreases considerably after age 50. Notably, closer investigation of the sleep classes reveals

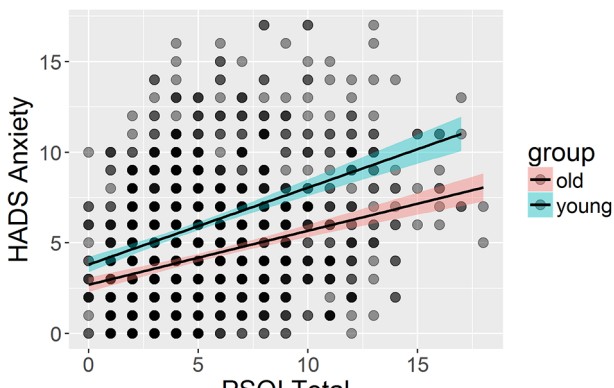

**Figure 6** Interaction between sleep quality and anxiety in the youngest third (n=723, age 18.48–46.2) compared with the oldest third of participants (n=724, age 71.79–98.88). HADS, Hospital Anxiety and Depression Scale; PSQI, Pittsburgh Sleep Quality Index.

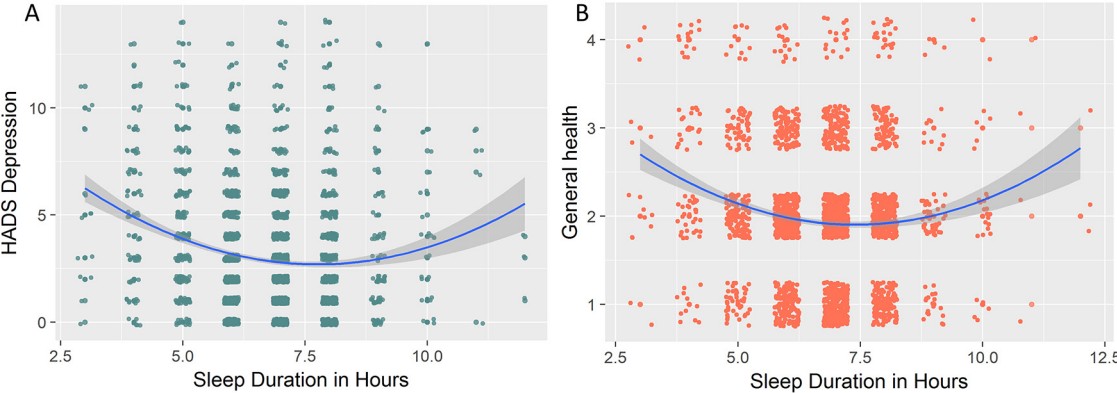

**Figure 7** Curvilinear associations between sleep duration in hours and (A) Hospital Anxiety and Depression Scale (HADS) depression and (B) general health (self-reported). For visual clarity, a small amount of random jitter was added to the data points.

likely further complexities of age-related differences. The category 'poor sleepers', most prevalent in older adults, shows high conditional likelihood of 'poor sleep' across all symptoms except 'daytime dysfunction'. One possible explanation is that almost all individuals in this group are beyond retirement age. For this reason, they likely have greater flexibility in tailoring their day-to-day activities to their energy levels (as opposed to individuals working full time) and are therefore less likely to consider themselves 'disrupted' even in the presence of suboptimal sleep. Although more detailed, interview-based investigations would be necessary to examine the precise nature of these findings, it stands to reason that certain symptoms change not just in prevalence but also in meaning across the lifespan.

One key strength of our broad phenotypic assessment is that it allows for direct comparison of the different measures of sleep quality and four key health domains. We found strongest associations between sleep quality and mental health, moderate relations between sleep quality and physical health and cognitive health and sleep, virtually all such that poorer sleep is associated with poorer health outcomes. We did not find evidence for associations between self-reported sleep and neural health. Notably, the relationships we observed are mostly stable across the lifespan, affecting younger and older individuals alike. A notable exception to these effects is the absence of any strong relation (after controlling for age) between sleep quality and neural health as indexed by tract-based average fractional anisotropy. Perhaps surprisingly, given the strong relationships in the same sample between sleep and other outcomes (eg, mental health, online supplementary figure 10), we found that self-reported sleep problems in a non-clinical sample are not associated with fractional anisotropy above and beyond old age. This is despite the fact that previous work within the same cohort observed moderate to strong associations between white matter and various cognitive outcomes.[42 80 81] However, although notable, our finding does not rule out that such associations do exist with

other white matter metrics, that they would be observed with objective measures of sleep such as polysomnography, or that the co-occurrence of age-related declines in sleep quality and white matter share an underlying causal association that cannot be teased apart in a cross-sectional sample.

One strength of our study is the assessment of neuroimaging metrics, namely, fractional anisotropy, in a large, community-dwelling healthy population. Fractional anisotropy is often used in studies of ageing (eg, Madden is relatively reliable[82]) and is sensitive to clinical anomalies such as WMHs. However, the relationship between FA and white matter health is indirect[40 83] and drawbacks include its inability to distinguish crossing fibres (eg, Jones et al[40] and Wandell[83]) and vulnerability to movement and the fact that it likely reflects a combination of underlying physiological properties. Various alternative white matter metrics exist, including summary measures of diffusivity (eg, axial/radial/mean diffusivity), volumetric measures of white matter hyperintensity and various innovative measures currently in development, but their physiological validity is ongoing.[83 84]

While there are limitations of self-report measures including in older cohorts,[19] including the fact that they likely reflect different aspects of sleep health than polysomnography (sleep in the lab), our results suggest that there are considerable advantages in using self-reported sleep measures: first, obtaining sleep quality data in a large and broadly phenotyped sample is feasible and second, our results demonstrated clear and consistent associations across multiple domains for both subjective (eg, self-reported health) and objective measures (eg, memory tests, BMI), which both replicate and extend previous lab-based sleep findings. Future work should ideally simultaneously measure polysomnography and self-report in longitudinal, large-scale cohorts to fully capture the range of overlapping and complementary relations between different aspects of sleep quality and health outcomes.[19]

For both self-report and objective measures of sleep quality, an open question is that of causality: does poor sleep affect health outcomes, do health problems affect sleep, are they both markers of some third problem or do causal influences go both ways? Most likely, all these patterns occur to varying degrees. Previous studies have shown that sleep quality causally affects health outcomes such as diabetes[43] and memory consolidation,[1] while other evidence suggests that depression directly affects sleep quality[85 86] and that damage to neural structures may affect sleep regulation.[87] Although our findings are in keeping with previous findings, our cross-sectional sample cannot tease apart the causal direction of the observed associations, more work remains to be done to disentangle these complex causal pathways.

In our paper, we focus on a healthy, age-heterogeneous community-dwelling sample. This allows us to study the associations between healthy ageing and self-reported sleep quality, but comes with two key limitations of the interpretations of our findings. First and foremost, our findings are cross-sectional, not longitudinal. This means that we can make inferences about age-related differences, but not necessarily age-related changes.[88 89] One reason why cross-sectional and longitudinal estimates may diverge is that older adults can be thought of as cohorts that differ from the younger adults in more ways than age alone. For example, our age range includes individuals born in the twenties and thirties of the 20th century. Compared with someone born in the 21st century, these individuals will likely have experience various differences during early life development (eg, less broadly accessible education, lower quality of healthcare, poorer nutrition and similar patterns). For some of our measures, these are inherent limitations—*truly* longitudinal study of neural ageing is inherently impossible as scanner technology has not been around sufficiently long. This means our findings likely reflect a combination of effects attributable to age-related changes as well as baseline differences between subpopulations that may affect both mean differences and developmental trajectories.

Second, our sample reflects an atypical population in the sense that they are willing and able to visit the laboratory on multiple occasions for testing sessions. This subsample is likely a more healthy subset of the full population, which will mean the range of (poor) sleep quality as well as (poorer) health outcomes will likely be less extreme than in the full population. However, this challenge is not specific to our sample. In fact, as the Cam-CAN cohort was developed using stratified sampling based on primary healthcare providers, our sample is likely as population representative as is feasible for a cohort of this magnitude and phenotypic breadth (see Shafto *et al*[12] for further details). Nonetheless, a healthier subsample may lead to restriction of range,[90] that is, an attenuation of the strength of the associations observed between sleep quality and health outcomes. Practically, this means that our results likely generalise to comparable, healthy community-dwelling adults, but not necessarily to populations that include those affected by either clinical sleep deprivation or other serious health conditions.

## CONCLUSIONS

Taken together, our study allows several conclusions. First, although we replicate the age-related deterioration in some aspects of sleep quality, other aspects remain stable or even improve. Second, we show that the profile of sleep quality changes across the lifespan. This is important methodologically, as it suggests that PSQI sum scores do not capture the full picture, especially in age-heterogeneous samples. Moreover, it is important from a psychological standpoint: we show that 'sleep quality' is a multidimensional construct and should be treated as such if we wish to understand the complex effects and consequences of sleep quality across the lifespan. Third, moderate to strong relations exist between sleep quality and cognitive, physical and mental health, and these relations largely remain stable across the lifespan. In contrast, we show evidence that in non-clinical populations, poorer self-reported sleep is not reliably associated with poorer neural health. Finally, we find that for absolute sleep duration, we replicate previous findings that both longer and shorter than average amounts of sleep are in association with poorer self-reported general health and higher levels of depression and anxiety.

Together with previous experimental and longitudinal evidence, our findings suggest that at least some age-related decreases in health outcomes may be due to poorer sleep quality. We show that self-reported sleep quality can be an important indicator of other aspects of healthy functioning throughout the lifespan, especially for mental and general physical health. Our findings suggest that accurate understanding of sleep quality is essential in understanding and supporting healthy ageing across the lifespan.

**Twitter** @rogierK

**Acknowledgements** We would like to thank Richard Morey and Eric-Jan Wagenmakers for valuable suggestions regarding the use of the BayesFactor package. We are grateful to the Cambridge Centre for Ageing and Neuroscience respondents and their primary care teams in Cambridge for their participation in this study. We also thank colleagues at the MRC Cognition and Brain Sciences Unit MEG and MRI facilities for their assistance.

**Collaborators** The Cam-CAN corporate author consists of the project principal personnel: Lorraine K Tyler, Carol Brayne, Edward T Bullmore, Andrew C Calder, Rhodri Cusack, Tim Dalgleish, John Duncan, Richard N Henson, Fiona E Matthews, William D Marslen-Wilson, James B Rowe; Research Associates: Karen Campbell, Teresa Cheung, Simon Davis, Linda Geerligs, Anna McCarrey, Abdur Mustafa, Darren Price, David Samu, Jason R Taylor, Matthias Treder, Kamen Tsvetanov, Janna van Belle, Nitin Williams; Research Assistants: Lauren Bates, Tina Emery, Sharon Erzinçlioglu, Sofia Gerbase, Stanimira Georgieva, Claire Hanley, Beth Parkin, David Troy; Affiliated Personnel: Tibor Auer, Marta Correia, Lu Gao, Emma Green, Rafael Henriques; Research Interviewers: Jodie Allen, Gillian Amery, Liana Amunts, Anne Barcroft, Amanda Castle, Cheryl Dias, Jonathan Dowrick, Melissa Fair, Hayley Fisher, Anna Goulding, Adarsh Grewal, Geoff Hale, Andrew Hilton, Frances Johnson, Patricia Johnston, Thea Kavanagh-Williamson, Magdalena Kwasniewska, Alison McMinn, Kim Norman, Jessica Penrose, Fiona Roby, Diane Rowland, John Sargeant, Maggie Squire, Beth Stevens, Aldabra Stoddart, Cheryl Stone, Tracy Thompson, Ozlem Yazlik; administrative staff: Dan Barnes, Marie Dixon, Jaya Hillman, Joanne Mitchell, Laura Villis.

**Contributors** AG, RK and MS designed the study. AG and RAK performed the analyses. CC organised and conducted the data collection. AG, MS and RAK wrote the manuscript. YL provided considerable expertise on sleep and poor sleep outcomes. All authors approved the final manuscript.

**Funding** The Cambridge Centre for Ageing and Neuroscience (Cam-CAN) research was supported by the Biotechnology and Biological Sciences Research Council (grant number BB/H008217/1). RAK is supported by the Sir Henry Wellcome Trust (grant number 107392/Z/15/Z) and by the UK Medical Research Council Programme (MC-A060-5PR60).

**Competing interests** None declared.

**Ethics approval** Ethical approval for the study was obtained from the Cambridgeshire 2 (now East of England-Cambridge Central) Research Ethics Committee (reference: 10/H0308/50).

**Provenance and peer review** Not commissioned; externally peer reviewed.

**Data sharing statement** The raw data and analysis code are available upon signing a data sharing request form (see http://www.mrc-cbu.cam.ac.uk/datasets/camcan/ for more detail).

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
