## [Reviewer comments · BMJ Open]

ARTICLE DETAILS

TITLE (PROVISIONAL)	How are age-related differences in sleep quality associated with health outcomes? An epidemiological investigation in a UK cohort of 2406 adults
AUTHORS	Kievit, Rogier; Gadie, Andrew; Shafto, Meredith; Leng, Yue; Cam-CAN, _

VERSION 1 - REVIEW

REVIEWER	Michael Scullin Baylor University, United States
REVIEW RETURNED	18-Nov-2016

GENERAL COMMENTS	Title: "Age-related differences in self-reported sleep quality predict healthy ageing across multiple domains: a multimodal cohort of 2406 adults" Authors: Gadie, Shafto, Leng, & Kievit Reviewed by: Michael Scullin Gadie et al. reported cross-sectional associations between aging, sleep, and health outcome variables. The strengths of the study included a large sample size (CAM-CAN project), inclusion and analysis of DTI white matter neuroimaging data, and inclusion of Bayesian analyses. I liked this paper overall and see value to its inclusion in the literature. A general theme, however, is that the authors may have attempted to do too much in this single manuscript and as a result the quality of each individual analysis was weaker than it could have been. I have listed below several recommendations for improving the manuscript. 1. The authors can bolster the rationale for their study. They should elaborate on why studying clinical samples is insufficient for understanding sleep—healthy associations beyond referring to simple generalizability. Some potential problems with only studying clinical populations is one might be introducing new mechanisms, third variables, etc., that cloud the relationship between sleep quality and the measured outcomes (of course, the authors may have additional concerns). Furthermore, the authors should specify in the rationale which specific health outcomes are being measured and why a multidomain approach is favorable to a focused approach. One potential limitation of a multidomain approach is that the quality of the literature review, methodology, and/or analysis might be weakened for each individual domain. Finally, there have been dozens (or more) of population-based studies using the PSQI and measuring cognitive, physical health, and mental health outcomes. The use of DTI is clearly the novel extension to this literature and should be highlighted accordingly.
---

2. The authors can be clearer as to whether the study is attempting to use a healthy sample or to include clinical groups. The authors reported that participants were not included if their MMSE was below 25 but the MMSE is not a good screening tool for mild dementia. The study's data showed large range of scores from 0 to 100 with a mean of 89 and standard deviation of 13. Those numbers are important because Mioshi et al. (2006) reported that a cutoff score of 88 had good sensitivity and specificity for dementia. What associations between sleep and cognition (or DTI) remain when eliminating participants whose ACE-R scores suggestive of dementia? Does eliminating those individuals with probable dementia lead to sleep—cognition/DTI interactions with chronological age?

3. The latent class analysis produced a very nice Figure 2. The rationale for the latent class analysis approach and its overall value to the paper were less clear. Why dichotomize responses and then categorize the participants when one can rely on a continuous measure? Furthermore, if one is to categorize the participants, why not rely on the PSQI's categorizations that were used for the correlational analyses? Moreover, why use any categorization when the raw continuous data (total sleep time and sleep onset latency) are available?

4. The treatment of sleep duration and blood pressure could be clarified. Most studies find an inverted U relationship between total sleep time and cognition, health, etc. It appears that the authors only tested linear relationships, but these variables warrant curvilinear tests. Long sleepers may be the individuals who show the poorest cognition and health. Grandner and Drummond (2007) wrote a good review article on long sleepers that may be helpful to the authors.

5. The authors stated that when controlling for age that the associations between sleep and cognitive variables were slightly attenuated. That was the case for verbal fluency, but almost all other measures in Supplementary Figure 3 showed strong evidence for the null hypothesis. Controlling for age is critical because it is well known that both sleep quality and cognition decline with age. My recommendation is for the authors to replace Figure 3 with the data in Supplementary Figure 3, or to place them side-by-side, and modify their interpretations accordingly.

6. I recommend caution in communicating that older adults who report poor sleep do not need to worry about brain health (p. 20, Lines 12-16). That may be true, but in the hands of the wrong news reporter that message might have unintended effects. Sleep and brain health associations might still be observed with other measures of sleep or with other measures of brain health.

Additional Comments:

Page 4, Lines 72-74. Polysomnography, not actigraphy, is the gold standard of sleep measurement.

In the Methods section, the authors referred the reader to published papers on the CAM-CAN protocol. That is fine for some details, but other details such as the timing of the scanning and cognitive tests relative to the PSQI are important.

In the Methods section, Line 117-118, make sure to cite the

	evidence that melatonin administration improves cognitive performance. Figure 2 – How do the authors interpret the daytime dysfunction findings? That value is low for all subtypes, but one would at least expect the poor sleepers (who seem to have every problem) would report daytime dysfunction. P. 15, Lines 46-53. Whether verbal fluency is sensitive to frontal lobe damage versus temporal lobe damage depends on type of task (phonemic or semantic). The authors included both types of verbal fluency tasks so include that the test may be sensitive to both frontal and temporal lobe damage. Fractional anisotropy is only one type of analysis of white matter. It has both strengths and weaknesses. The potential limitations of the FA approach should be described in the discussion section.
--	--

REVIEWER	Dr Marta Jackowska Roehampton University, London, UK
REVIEW RETURNED	09-Dec-2016

GENERAL COMMENTS	BMJ Open Age-related differences in self-reported sleep quality predict healthy ageing across multiple domains: a multi-modal cohort of 2406 adults. This study reports on associations between self-reported sleep and 4 outcomes (physical health, cognitive health, mental health and neural health) in a sample of adults aged 18-98 years. As can be expected, given extant literature, poor global sleep quality was associated with worst mental health, less favourable cognitive function and physical health (BMI, blood pressure) but there was no link between sleep and neural health. These associations did not seem to be moderated by age. Abstract Given this study is based on cross-sectional data it is not correct to say in the Abstract's objectives that this study aimed "To examine lifestyle changes in self-reported sleep quality....". This should be rephrased. Background l.72, p.4 "...actigraphy (measuring sleep quality in the lab) is commonly considered the gold standard" –this sentence is erroneous, actigraphy is an ambulatory sleep measure where sleep can be measured objectively without the need to stay in a sleep lab. Polysomnography is measured in a sleep lab and is the gold standard sleep measure. The same mistake is made in the Discussion. l. 86, p.5 I am not sure how novel it is to look at the relationship between sleep quality and health (e.g. BMI, blood pressure) given there are a number of reviews and meta-analyses of prospective studies looking at these health outcomes and sleep. Methods p. 5 the sample is not described at all, in particular it is not clear how many older people were in this sample, some basic socio-demographic information should also be given despite the fact that
--

	this cohort's details have been published elsewhere. Results The results are clearly written and informative but the large number of figures is rather confusing ; for example I am not sure how useful all these figures and tables will be for a sleep clinician, or a medical doctor and/or GP. I.16, p.14 the sentence starting with "Next" does not make sense. Discussion I find it puzzling that despite the authors' aim to look at age-related changes in sleep and the relationship between sleep and health outcomes no reference is made to the literature that has looked into age-related changes in sleep, e.g. Ohayon MM, Carskadon MA, Guilleminault C, Vitiello MV. Meta-analysis of quantitative sleep parameters from childhood to old age in healthy individuals: developing normative sleep values across the human lifespan. Sleep 2004; 27: 1255-1274. Crowley K. Sleep and sleep disorders in older adults. Neuropsychol Rev 2011; 21: 41-53. Because of that, having read this rather long paper, I am not sure what novel this study actually finds, or what is the "take home message" from this paper.
--	---

VERSION 1 – AUTHOR RESPONSE

Reviewer: 1

Reviewer Name: Michael Scullin

Institution and Country: Baylor University, United States

Please state any competing interests: None declared

Please leave your comments for the authors below

Title: "Age-related differences in self-reported sleep quality predict healthy ageing across multiple domains: a multimodal cohort of 2406 adults"

Authors: Gadie, Shafto, Leng, Cam-CAN & Kievit

Reviewed by: Michael Scullin

'1. The authors can bolster the rationale for their study. They should elaborate on why studying clinical samples is insufficient for understanding sleep—healthy associations beyond referring to simple generalizability. Some potential problems with only studying clinical populations is one might be introducing new mechanisms, third variables, etc., that cloud the relationship between sleep quality and the measured outcomes (of course, the authors may have additional concerns).'

We thank the reviewer for their thoughtful comments. We agree our manuscript is ambitious and that a clearer rationale is essential to allow readers to synthesize the large set of analyses. We have further clarified these points in our introduction (page 5, lines 81-92).

'Furthermore, the authors should specify in the rationale which specific health outcomes are being measured and why a multidomain approach is favorable to a focused approach. One potential limitation of a multidomain approach is that the quality of the literature review, methodology, and/or

analysis might be weakened for each individual domain.'

We thank the reviewer for raising this point. Although a more focused investigation of individual bivariate associations can be valuable, especially for the reasons the reviewer outlines, our approach of examining multiple domains simultaneously has other strengths, now more clearly emphasized in the introduction. By investigating associations with different domains in the same sample, we can make direct comparisons of the relative magnitudes of these effects – This is much harder when individual studies report effects in smaller, often idiosyncratic cohorts ranging from clinical to healthy. However, to ameliorate these concerns we have emphasized our rationale along the above lines in the introduction.

'Finally, there have been dozens (or more) of population-based studies using the PSQI and measuring cognitive, physical health, and mental health outcomes. The use of DTI is clearly the novel extension to this literature and should be highlighted accordingly.'

We thank the reviewer for highlighting the novelty of the large sample DTI analysis. We have emphasized this unique aspect of our study more in both the introduction and the discussion.

'2. The authors can be clearer as to whether the study is attempting to use a healthy sample or to include clinical groups. The authors reported that participants were not included if their MMSE was below 25 but the MMSE is not a good screening tool for mild dementia. The study's data showed large range of scores from 0 to 100 with a mean of 89 and standard deviation of 13. Those numbers are important because Mioshi et al. (2006) reported that a cutoff score of 88 had good sensitivity and specificity for dementia. What associations between sleep and cognition (or DTI) remain when eliminating participants whose ACE-R scores suggestive of dementia? Does eliminating those individuals with probable dementia lead to sleep—cognition/DTI interactions with chronological age?'

We thank the reviewer for this suggestion, and agree that to some extent any cut off criterion is inherently arbitrarily. However, we note that the cut off we employed is also relatively common, especially in cognitive neuroimaging. Moreover, it is not entirely clear why we would necessarily want to implement a more stringent cut off – All individuals in our cohort are as closely population representative of a general healthy population, which will also include individuals in the (very) early stages of mild dementia. Nonetheless, we implemented the different cut off the reviewer suggested and reran all analyses (explicitly mentioning both exclusion criteria of Pg5). The broad findings are very similar, although we highlight the most relevant differences in the manuscript.

Note that other, more stringent cut-offs are sometimes employed to screen for premorbid dementia, such as a score of 88 or higher in the Addenbrookes Cognitive Examination – Revised (27). For the sake of comprehensiveness we repeated our analyses using this more stringent cut off (ACE-R>88), but observed no noteworthy differences in our findings, so we only report the findings based on the MMSE.

'3. The latent class analysis produced a very nice Figure 2. The rationale for the latent class analysis approach and its overall value to the paper were less clear. Why dichotomize responses and then categorize the participants when one can rely on a continuous measure? '

We agree with the reviewer that our analysis could be perceived as not taking into account the full range of symptom space. Our intention was to emphasize our finding (using regression analyses) that conceptualizing 'poor sleep' as a single dimension does not reflect the subtleties in lifespan differences. This is reflected in the fact that the PSQI sum score only captures a small part of the age-related differences, as well as in the distinct latent class profiles we observe. In other words, our conceptual point was to get across that not all poor sleep is created equally, and that standards use of

certain metrics might obscure specific profiles of sleep problems present even in non-clinical populations. We agree with the reviewer that the justification for our LCA was not as clear as it could have been – We have expanded this accordingly in our revision (line 236 and on).

Secondly, statistically our measures of PSQI symptoms straddle the border between continuous and categorical – Although some are fully continuous (e.g. sleep latency) others are less so. For instance, although scored on a range of four several of the scales (such as Subjective Sleep quality) have implicitly binary response options of ‘Very good’ and ‘fairly good’ on the one hand and ‘fairly bad’ and ‘very bad’ on the other – We would argue these represent something between fully continuous and fully categorical. More importantly, analytical work in psychometrics (Rhemtulla et al., 2012) suggests that likert-like graded scales can be treated as continuous only from five ordinal categories upwards – Given that we have a four scale outcome, and one that contains implicit binary divisions, by fitting an LCA we are erring on the side of caution (rather than fitting, say, a latent profile analysis, which would likely give similar results). Finally, although our analysis divides individuals into discrete classes with specific profiles, it is still possible to examine the conditional response likelihood of responding ‘yes’ to each symptom as a continuous metric (between 0 and 1) that reflects the nature of the association between the class and the outcome, so to some degree the continuous nature of the symptoms is (partly) preserved. To summarize: Our psychometric analysis treats the variables as categorical based on analytical recommendations and to err on the side of caution. To ameliorate these reasonable concerns, we have added this justification more explicitly in the manuscript.

Rhemtulla, M., Brosseau-Liard, P. É., & Savalei, V. (2012). When can categorical variables be treated as continuous? A comparison of robust continuous and categorical SEM estimation methods under suboptimal conditions. *Psychological methods*, 17(3), 354. Chicago

'Furthermore, if one is to categorize the participants, why not rely on the PSQI's categorizations that were used for the correlational analyses?'

We are not 100% sure what the reviewer is suggesting in terms of PSQI categorization? If they mean to classify individuals on each individual symptom, this is certainly possible but would miss the richness of multivariate classifier. For instance, in our multiple regression we already show that the PSQI sumscores change very little across the lifespan, but the multivariate pattern of individual symptoms does change considerably. For this reason, using all individual PSQI scales simultaneously provides a richer picture of age-related differences, as exemplified in Figure 2 – In our view this provides new insight into age-related differences in sleep dynamics, and the classes we observe offer a relatively simple to understand summary of certain types of sleep problems that would be hard to view based only on the individual scales. Note that if this was unclear: the latent class analysis uses the same data as the regression analyses, but dichotomized to abide by distributional assumptions.

'Moreover, why use any categorization when the raw continuous data (total sleep time and sleep onset latency) are available?'

For three reasons as outlined above: Firstly, the majority of the metrics straddle the boundary between continuous and ordinal. Secondly, if we restrict ourselves only to the subset of two measures outlined above, our picture of sleep differences across the lifespan would be much more impoverished than our current analysis across all symptoms. Third, in our view this categorization (although inherently an oversimplification) makes certain patterns (much) easier to interpret. We hope this alleviates the reviewers concerns.

'4. The treatment of sleep duration and blood pressure could be clarified. Most studies find an inverted U relationship between total sleep time and cognition, health, etc. It appears that the authors only tested linear relationships, but these variables warrant curvilinear tests. Long sleepers may be the

individuals who show the poorest cognition and health. Grandner and Drummond (2007) wrote a good review article on long sleepers that may be helpful to the authors.'

We agree with the reviewer, and are aware of the observations regarding inverted U-shapes. Our consideration was that we already have such an extended set of analysis that non-linear approaches would further extend our already lengthy manuscript. However, upon reflection we agree that, especially in such a large sample, it is important to include this analysis, so we have done so under a new header at the end of the results. We observe evidence for u-shaped associations with general health, health over the last 12 months and anxiety and depression, but not for any cognitive or neural outcomes. We have included two figures for the most striking associations and copy the new section below.

Non-linear associations between sleep and health outcomes

In the above analyses, we focused on linear associations between symptoms and health outcomes. However, for one aspect of sleep, namely sleep duration (in hours), evidence exists that these associations are likely to be non-linear, such that both shorter and longer than average sleep are associated with poorer health outcomes (e.g. 84–86). This is echoed in clinical criteria for depression, which commonly include that include both hyper- and hypo-somnia as 'sleep disruption' symptoms – In other words, both too much or too little sleep are suboptimal. To examine whether we observe evidence for non-linearities we examined the relationship between raw scores on sleep duration (in hours, not transformed to PSQI norms) and health outcomes across the four domains. If the association between sleep and outcomes is indeed u-shaped (or inverted U, depending on the scale) then a Bayesian regression would prefer the less parsimonious model that includes the quadratic term. We observed no non-linear associations between any neural or cognitive health variables. We find strong evidence for a quadratic (subscript q) over a linear (subscript l) associations between sleep duration and HADS anxiety ($\log\text{BF}_{q|l} = 19.98$), even more strongly so with HADS Depression ($\log\text{BF}_{q|l} = 25.83$, see Figure 7A shows the strongest curvilinear association, namely with depression). We find a similar u-shaped curve with general health ($\text{BF}_{q|l} = 277.81$) and self-reported health over the last 12 months ($\text{BF}_{q|l} = 887.59$), the latter shown in Figure 7b. Together, these analyses support previous conclusions that some (although not all) poorer health outcomes can be associated with both too much and too little sleep.

'5. The authors stated that when controlling for age that the associations between sleep and cognitive variables were slightly attenuated. That was the case for verbal fluency, but almost all other measures in Supplementary Figure 3 showed strong evidence for the null hypothesis. Controlling for age is critical because it is well known that both sleep quality and cognition decline with age. My recommendation is for the authors to replace Figure 3 with the data in Supplementary Figure 3, or to place them side-by-side, and modify their interpretations accordingly.'

We appreciate the point by the reviewer and agree. We have altered the Results accordingly, switching the Supplementary figure 3 and figure 3 accordingly.

'6. I recommend caution in communicating that older adults who report poor sleep do not need to worry about brain health (p. 20, Lines 12-16). That may be true, but in the hands of the wrong news reporter that message might have unintended effects. Sleep and brain health associations might still be observed with other measures of sleep or with other measures of brain health.'

We agree with the reviewer that our phrasing was too strong and could easily be misinterpreted. We have rephrased as follows:

Perhaps surprisingly, given we found strong relationships in the same sample between sleep and other outcomes (e.g. mental health, Figure 10) we find that self-reported sleep problems in a non-

clinical sample are not associated with fractional anisotropy above and beyond old age. This is despite the fact that previous work within the same cohort observed moderate to strong associations between white matter and various cognitive outcomes (40,85,86). However, although notable, our finding does not rule out that such associations do exist with other white matter metrics, that they would be observed with objective measures of sleep such as polysomnography, or that the co-occurrence of age-related declines in sleep quality and white matter share an underlying causal association that cannot be teased apart in a cross-sectional sample.

Additional Comments:

'Page 4, Lines 72-74. Polysomnography, not actigraphy, is the gold standard of sleep measurement.'

We thank the reviewer for correcting our mistake. We have updated accordingly throughout the manuscript.

'In the Methods section, the authors referred the reader to published papers on the CAM-CAN protocol. That is fine for some details, but other details such as the timing of the scanning and cognitive tests relative to the PSQI are important.'

We agree that such details should be included, and have updated our manuscript with appropriate details, including

- Under "Methods; Sample" new information has been inserted between lines 107 and on regarding participant sampling, dates and locations
- The timing of the DWI acquisition has been included on Page 6

'In the Methods section, Line 117-118, make sure to cite the evidence that melatonin administration improves cognitive performance.'

We have now included relevant citations supporting this statement, and thank the reviewers for pointing out this omission.

'Figure 2 – How do the authors interpret the daytime dysfunction findings? That value is low for all subtypes, but one would at least expect the poor sleepers (who seem to have every problem) would report daytime dysfunction.'

We agree this is an initially counterintuitive finding. Our speculative interpretation, which we have now added to the manuscript, is that this variable is intrinsically linked to the nature of day to day activities. The poor sleepers are most frequent in (very) advanced age – Virtually all individuals in this group are beyond retirement age. For this reason, they likely have greater flexibility in tailoring their day to day activities to their energy levels (as opposed to individuals working fulltime), and are therefore less likely to consider themselves 'disrupted' even in the presence of suboptimal sleep.

'P. 15, Lines 46-53. Whether verbal fluency is sensitive to frontal lobe damage versus temporal lobe damage depends on type of task (phonemic or semantic). The authors included both types of verbal fluency tasks so include that the test may be sensitive to both frontal and temporal lobe damage.'

We thank the reviewer for their suggestion. We have now clarified (pg 16, lines 23-24) that the neuropsychological test is sensitive to both frontal and temporal lobe damage, and have provided a relevant article discussing these region specific anatomical correlates in greater depth.

'Fractional anisotropy is only one type of analysis of white matter. It has both strengths and

weaknesses. The potential limitations of the FA approach should be described in the discussion section.'

We agree with the reviewer that FA is only one measure, and like all such measures is imperfect. We have added the below segment to the limitations section.

One strength of our study is the assessment of neuroimaging metrics, namely fractional anisotropy, in a large, community-dwelling healthy population. Fractional anisotropy is often used in studies of aging (e.g. Madden, is relatively reliable (87)) and is sensitive to clinical anomalies such as white matter hyperintensities. However, the relationship between FA and white-matter health is indirect (88,89) and drawbacks include its inability to distinguish crossing fibers (e.g. (80) and vulnerability to movement and the fact that it likely reflects a combination of underlying physiological properties. Various alternative white matter metrics exist, including summary measures of diffusivity (e.g. axial/radial/mean diffusivity), volumetric measures of white matter hyperintensity (e.g.) and various innovative measures currently in development, but their physiological validity is ongoing (88,90).

Reviewer: 2

Reviewer Name: Dr Marta Jackowska

Institution and Country: Roehampton University, London, UK

Please state any competing interests: None declared

BMJ Open Age-related differences in self-reported sleep quality predict healthy ageing across multiple domains: a multi-modal cohort of 2406 adults. This study reports on associations between self-reported sleep and 4 outcomes (physical health, cognitive health, mental health and neural health) in a sample of adults aged 18-98 years. As can be expected, given extant literature, poor global sleep quality was associated with worst mental health, less favourable cognitive function and physical health (BMI, blood pressure) but there was no link between sleep and neural health. These associations did not seem to be moderated by age.

'Abstract Given this study is based on cross-sectional data it is not correct to say in the Abstract's objectives that this study aimed "To examine lifestyle changes in self-reported sleep quality...." . This should be rephrased. '

We agree that our language use was imprecise, and have rephrased throughout the manuscript to age related differences instead of changes

'Background I.72, p.4 "...actigraphy (measuring sleep quality in the lab) is commonly considered the gold standard" –this sentence is erroneous, actigraphy is an ambulatory sleep measure where sleep can be measured objectively without the need to stay in a sleep lab. Polysomnography is measured in a sleep lab and is the gold standard sleep measure. The same mistake is made in the Discussion. I. 86, p.5 '

We thank the reviewer for correcting our mistake. We have updated accordingly throughout the manuscript.

'I am not sure how novel it is to look at the relationship between sleep quality and health (e.g. BMI, blood pressure) given there are a number of reviews and meta-analyses of prospective studies looking at these health outcomes and sleep.'

The novelty in our manuscript comes in three forms: First, we examine the association between sleep

and white matter health in a large, healthy community dwelling sample, where most neuroimaging work focuses on smaller samples displaying some form of sleep pathology. Second, we have examined the associations between sleep quality and health across four domains, including neural health, simultaneously within the same population. This allows us, in contrast to most if not all other papers, to compare the magnitude of associations across domains within the same population. Third, we use innovative methodology that allows us to quantify the presence and absence of associations, something traditional NHST methods cannot do. We have emphasized these novel aspects of our manuscript in the introduction (lines 78-90) and abstract, and more briefly summarized the take home messages.

'Methods p. 5 the sample is not described at all, in particular it is not clear how many older people were in this sample, some basic socio-demographic information should also be given despite the fact that this cohort's details have been published elsewhere.'

We agree we could have been clearer here – We have addressed this in the revision with further protocol details

'Results The results are clearly written and informative but the large number of figures is rather confusing ; for example I am not sure how useful all these figures and tables will be for a sleep clinician, or a medical doctor and/or GP. '

Our paper tries to strike a balance between comprehensiveness and intelligibility. Although we don't want to compromise on the amount of available detail to ensure other researchers are able to replicate and properly interpret our findings, we have improved our summary of our key findings.

'l.16, p.14 the sentence starting with "Next" does not make sense.'
Now fixed

'Discussion I find it puzzling that despite the authors' aim to look at age-related changes in sleep and the relationship between sleep and health outcomes no reference is made to the literature that has looked into age-related changes in sleep, e.g. Ohayon MM, Carskadon MA, Guilleminault C, Vitiello MV. Meta-analysis of quantitative sleep parameters from childhood to old age in healthy individuals: developing normative sleep values across the human lifespan. *Sleep* 2004; 27: 1255-1274. Crowley K. Sleep and sleep disorders in older adults. *Neuropsychol Rev* 2011; 21: 41-53. '

Although given the length of our manuscript we aimed to keep the introduction relatively condensed, we agree with the reviewer that further background on general sleep and ageing patterns are important. We thank the reviewer for highlighting these articles – We agree they are highly relevant and have incorporated them in the introduction accordingly.

'Because of that, having read this rather long paper, I am not sure what novel this study actually finds, or what is the "take home message" from this paper.'

The novelty in our manuscript comes in three forms: First, we examine the association between sleep and white matter health in a large, healthy community dwelling sample, where most neuroimaging work focuses on smaller samples displaying some form of sleep pathology. Second, we have examined the associations between sleep quality and health across four domains, including neural health, simultaneously within the same population. This allows us, in contrast to most if not all other papers, to compare the magnitude of associations across domains within the same population. Third, we use innovative methodology that allows us to quantify the presence and absence of associations, something traditional NHST methods cannot do. We have emphasized these novel aspects of our manuscript in the introduction and abstract, and more briefly summarized the take home messages.

VERSION 2 – REVIEW

REVIEWER	Michael Scullin Baylor University, USA
REVIEW RETURNED	03-Mar-2017

GENERAL COMMENTS	My comments have been addressed.
----------------------------------

REVIEWER	Dr Marta Jackowska Roehampton University, London, UK
REVIEW RETURNED	03-Mar-2017

GENERAL COMMENTS	The authors have considerably improved the quality of the article; in particular I was pleased to see much clear and bolstered study aims and take home message. Thank you. All my comments have been well addressed.
---